# C-Terminal Modification on the Immunomodulatory Activity, Antioxidant Activity, and Structure–Activity Relationship of Pulsed Electric Field (PEF)-Treated Pine Nut Peptide

**DOI:** 10.3390/foods11172649

**Published:** 2022-08-31

**Authors:** Shuyu Zhang, Liu Dong, Zhijie Bao, Songyi Lin

**Affiliations:** National Engineering Research Center of Seafood, School of Food Science and Technology, Dalian Polytechnic University, Dalian 116034, China

**Keywords:** modified peptide, macrophage activation, antioxidant activity, pulsed electric field, secondary structure

## Abstract

In this study, a novel peptide VNAVL was synthesized by removing the C-terminal histidine on the basis of a bioactive peptide VNAVLH obtained from pine nut (Pinus koraiensis Sieb. et Zucc) protein. The effects of removing histidine on antioxidant activity, immunomodulatory activity, and secondary structure of the PEF-treated peptide were discussed. Compared with VNAVLH, VNAVL only exhibited lower antioxidant activity, but no immunomodulatory activity to release TNF-α, IL-6, and NO by activating RAW 264.7 cells. In addition, both antioxidant and immune activities of VNAVLH were significantly more sensitive to treatment with 40 kV/cm than other field intensities, whereas VNAVL was not sensitive to field strength changes. CD spectra and DSSP analysis verified that both peptides consisted of a β structure and random coil, but the ability of VNAVL to transform the random coil via PEF treatment is weaker than that of VNAVLH. Therefore, PEF treatment might expose the key active site located on the C-terminal histidine by altering the secondary structure of the peptide.

## 1. Introduction

Food-derived bioactive peptides are intriguing constituents for use as functional foods and nutraceuticals due to their advantageous qualities in antibacterial, antihypertensive, antioxidant, and immunomodulatory peptide activities [1,2,3,4]. Significant research has been conducted to understand the structure–activity relationship of bioactive peptides in order to develop functional foods and nutraceuticals that are based on these potential bioactive peptides [5]. The relationship between structure and activity is frequently investigated from the primary structure, hydrophilicity, amino acid compositions, net charge, secondary structures, and spatial configurations [6]. However, the majority of current research focuses on the structure–activity relationship of native peptides. Chemical modifications to peptide structures show great potential to affect their biological activities and physiological characteristics [7]. Amino acid residues of the peptides can be further glycated, such as with GMP, or as a result of the Maillard process, phosphorylated, such as with casein phosphopeptides or phosvitin peptides, or cyclic, such as with flaxseed cyclolinopeptides. Peptide modifications can happen readily during food preparation or as a result of interactions with other food components [8].

Recently, the extraction and purification of bioactive compounds have employed PEF treatment, a potential non-thermal treatment method [9,10,11]. It is notable that PEF treatment might improve peptide biological activity in vitro under the appropriate conditions, and the optimal conditions are mostly in the range of 10-40 kV/cm [12,13,14,15,16]. In addition, the mechanisms of improvement of the biological activity of food-derived peptides by PEF were also investigated in recent studies. Zhang et al. illuminated the possible mechanism of PEF treatment on the antioxidant activity of the pine nut peptide QWFH [17]. In ideal circumstances, PEF treatment could promote the antioxidant activities of pine nut peptides, which were mainly associated with the secondary structure of peptides. Zhang et al. demonstrated the influence of PEF on RGAVLH by using macrophage phagocytic activity and NO production as immunomodulatory assessment indices [18]. The phagocytic ability of peptides was increased by 12.09%, and NO production increased by 65.00%, and the improvement of immunomodulatory activity was found to be caused by the change in steric state and steric force in the peptide solution system. Although improvements in the biological activity of food-derived peptides by PEF treatment have been previously studied in relation to the secondary structure, there is limited research on the structural modification of peptides via PEF treatment.

In previous work, the peptide VNAVLH was purified through Sephadex gel filtration chromatography from pine nut protein hydrolysate, and analyzed via HPLC-MS/MS installed with a C18 column [19]. We discovered that the antioxidant activity of VNAVLH can be enhanced by PEF treatment, and preliminarily speculated that the C-terminal amino acid histidine might be the action site. In the current research, a novel peptide VNAVL was designed and synthesized by removing the histidine at the C-terminus of the original peptide. The effects of PEF treatment on the antioxidant properties of the peptides were ascertained using DPPH and ABTS radical scavenging assays. The immunomodulatory effects of PEF treatment on peptides were assessed by RAW 264.7 cells, and the impacts of the secondary structure on peptides were explored using CD and DSSP analysis via MD simulation. In this study, we explored the impacts of PEF therapy on the relationship between the structure and activity of peptides and the underlying mechanisms, and further confirmed the key amino acid sites of PEF treatment on the original peptide sequence.

## 2. Material and Methods

### 2.1. Materials and Reagents

The pine nut peptide Val-Asn-Ala-Val-Leu-His (VNAVLH) and C-terminal-modified peptide Val-Asn-Ala-Val-Leu (VNAVL) were obtained from Hefei Saimanuo Biological Technology Co., Ltd. (Hefei, Anhui Province, China). The purity of the peptides was 99.40% and 99.34%, and the molecular weights were 651.77 and 514.61 Da, respectively. The RAW 264.7 cell line was acquired from the type culture collection of the Chinese academy of sciences (Shanghai, China). DPPH, ABTS, MTT, and lipopolysaccharides (LPS) were acquired from Sigma Chemicals Co. (Saint Louis, MO, USA). Mouse TNF-α and IL-6-detecting ELISA kits were acquired from Solarbio Science & Technology Co. (Beijing, China). The NO assay kit and Neutral red were acquired from Beyotime Biotechnology Co. (Shanghai, China). Other reagents were acquired from Gibco BRL Life Technology (Gaithersburg, MD, USA).

### 2.2. PEF Treatment on Peptides

The PEF treatment method for peptides was modified from the method of Lin et al. [20]. The peptides were dissolved in deionized water (8 mg/mL). Then, the pipes were cleaned with deionized water and absolute ethanol, respectively. Then, 3.2 mL/min of the peptide solution was injected into the PEF system at a temperature of 25 °C. Electric field intensities were set from 10 kV/cm to 40 kV/cm, with the frequency remaining constant at 2000 Hz. For use in experiments, the appropriate peptide solution (8 mL) was gathered.

### 2.3. Determination of DPPH Radical Scavenging

The DPPH radical scavenging experiment was carried out as described by Yang et al. with a few minor adjustments [21]. In 96-well plates, 100 μL of the peptide solution (2 mg/mL) was combined with 100 μL of the DPPH (0.6 mM) solution and 100 μL of the methanol solution. The plates were then incubated for 30 min. Then, 100 μL of DPPH and 200 μL of methanol made up the blank group. Using a microplate reader, absorbance at 515 nm was calculated. The following equation was used to determine the DPPH radical scavenging capacity:$$\mathrm{DPPH}\;\mathrm{radical}\;\mathrm{scavening}\;\mathrm{activity}\;\left(\%\right)=1-\frac{Asample}{Ablank}\times 100\%$$

### 2.4. Determination of ABTS Radical Scavenging

The procedure outlined by Tironi et al. was followed while performing the ABTS radical scavenging assay [22]. Potassium persulfate (2.45 mM) was mixed with the ABTS (7 mM) solution, which was then left at room temperature for 15 h while being shielded from light. Deionized water was used to dilute the ABTS solution prior to the experiment to attain an absorbance of 0.70 ± 0.02 at 734 nm. In 96-well plates, 50 microliters of the peptide solution (2 mg/mL) were mixed with 100 μL of the diluted ABTS solution in a 96-well plate and then incubated for 6 min in the dark. Then, 50 L of deionized water and 100 L of the diluted ABTS solution made up the blank group. Using a microplate reader, absorbance at 734 nm was calculated. The following formula was used to calculate ABTS’s scavenging activity:$$\mathrm{ABTS}\;\mathrm{radical}\;\mathrm{sacvening}\;\mathrm{activity}\;\left(\%\right)=1-\frac{Asample}{Ablank}\times 100\%$$

### 2.5. Determination of Phagocytosis Assay in RAW 264.7 Cells

Phagocytosis activity is one of the most significant indicators for evaluating immune toxicity and changes in macrophage function, as it represents the initial and most essential defense role of macrophages in immunological response [23]. Bioactive peptides can bind to pattern recognition receptors (such as TLRs) on the surface of macrophages, and then initiate related signaling pathways to further activate macrophages [24]. The influence of peptides on macrophage phagocytic activity was examined using the neutral red phagocytosis assay [25]. After culture, 10^5^ RAW 264.7 cells per well in a 96-well plate were exposed to the samples treated by PEF for 24 h. LPS was used as a positive control, whereas the peptide was replaced with a full medium as a blank control. The extra neutral resolution was then removed by washing the cells three times in PBS. After that, each well received 100 μL of celllysate (1 M anhydrous ethanol: acetic acid 1:1, *v/v*) before the cells were lysed for 30 min. With the use of a microplate reader, the absorbance at 540 nm was measured. The phagocytosis index was calculated using the formula below:$$\mathrm{phagocytosis}\;\mathrm{index}=\frac{Asample-Acontrol}{Acontrol}\times 100\%$$

### 2.6. Determination of NO, TNF-α and IL-6 Production in RAW 264.7 Cells

At 37 °C, RAW 264.7 cells were incubated in a humid environment with 5% CO_2_. A 96-well plate was loaded with cells, adjusted to 1 × 10^6^ cells/mL, and incubated for 24 h. The cells were then exposed to untreated and PEF-treated samples and incubated for an additional 24 h. LPS was used as a positive control, whereas the peptide was replaced with the full medium as a blank control. MTT reagent was used to assess the toxicity of the peptides to cells. The levels of NO, TNF, and IL-6 in the cell supernatants were determined by an ELISA kit.

### 2.7. CD Spectroscopy Assay

The secondary structure of the peptides was discovered using a Jasco J-810 CD Spectrometer (Shimadzu, Kyoto, Japan) [26]. Nitrogen (20 L/min) was used to clean the CD spectroscopy for 20 min prior to measurement. At a temperature of 25 °C, the CD wavelength range was set at 190–240 nm, with a scanning speed of 20 nm/min. The resolution, slit width, and optical path length were all set to 0.2 nm, 0.1 cm, and 2 nm, respectively. The concentration of the untreated and PEF-treated peptide solutions was 0.05 mg/mL for the determination of the CD. The secondary structure of the peptide was calculated by CDPro software (Warsaw, Poland) with the following equation:$$\left[\theta \right]=\theta /10LC_{M}N$$
where *θ* denotes the CD value acquired from the apparatus, *L* denotes the path length (cm), *C_M_* denotes the peptide concentration (mol/L), and *N* denotes the total number of amino acid residues in the samples. The secondary structure curve can be obtained from [*θ*]. The percent secondary structure composition of each peptide was calculated by Spectra Manager using the Selcon3 program. Although the procedure is empirical, the relative content of the generated secondary structure is valid. Therefore, the data obtained using this method is essentially consistent with the facts.

### 2.8. Molecule Dynamics Simulation

The GROMACS package 2016.4 was used to run the MD simulations in the CHARMM36 all-atom force field [27]. Discovery Studio 4.5 software (Accelrys Co., Ltd., San Diego, CA, USA) created the first peptide structure of VNAVLH and VNAVL, and GROMACS tools were used to analyze and visualize molecular movement trajectories (version 1.9.3) (China’s Sugon Co., Ltd, Tianjin, China.) [28]. The systems were dismantled using counterions of Na+ and Cl^−^. To prevent system breakdowns, 5000 steps of energy minimization using the conjugated gradient technique were performed prior to the MD simulations with a convergence criterion of 100.00 kcal/mol Å^−1^ [29]. Trajectories of long-playing MDs were investigated using the GROMACS plugin. The secondary structure was calculated by the built-in tool do_dssp [30].

### 2.9. Statistical Analysis

All tests were performed in at least triplicate, the results were expressed as an average with standard deviation (SD), and the data were used for one-way ANOVA using Origin Pro 8.0 (OriginLab, Corvallis, OR, USA) and SPSS 13.0 (IBM, New York, NY, USA), respectively. *p* < 0.05 was used to evaluate significance. *R*^2^ is a measure of how well the polynomial model equation fits.

## 3. Results and Discussion

### 3.1. Antioxidant Activity of Peptides

In the current research, the original peptide (VNAVLH) and C-terminal modified peptide (VNAVL) were treated by PEF with different electric field intensities. The effects of PEF treatment on the antioxidant capacity of the peptides are depicted in Figure 1. Both DPPH and ABTS scavenging activity of two peptides were significantly improved after PEF treatment, and the antioxidant activities of VNAVLH were significantly stronger than that of VNAVL before and after PEF treatment (*p* < 0.05). Compared with the untreated sample, the DPPH scavenging activity of VNAVLH were significantly (*p* < 0.05) increased by 6.50 ± 0.31%, 6.56 ± 0.06%, 7.31 ± 0.43%, and 9.77 ± 0.19%, respectively. It was notable that DPPH scavenging activity of VNAVLH at 40 kV/cm was significantly higher (*p* < 0.05) than the other field intensities, which was not the case for VNAVL, and there was no significant difference (*p* > 0.05) in VNAVL under different field intensity treatments (Figure 1A). Additionally, the ABTS scavenging activity of VNAVLH at 20 kV/cm (26.38 ± 0.23%), 30 kV/cm (26.96 ± 0.50%), and 40 kV/cm (27.75 ± 0.23%) was significantly (*p* < 0.05) higher than 10 kV/cm (23.98 ± 0.34%), while there was also no significant difference (*p* > 0.05) in VNAVL under different field intensity treatments, as shown in Figure 1B.

The results indicate that both the original and modified peptides had antioxidant activity, but the antioxidant activity of the modified peptide was much lower than that of the original peptide, which might be due to the proton-donating ability of the imidazole group of histidine in the peptide [31]. Moreover, the original peptide was substantially more sensitive to the 40 kV/cm field intensity treatment than the other field intensities, which was absent from the C-terminal-modified peptide. It might be due to the stronger electric field intensity (40 kV/cm) generated by the positive and negative electrodes of the PEF device, so that the cation of the only positively charged histidine leaves the peptide surface and provides more protons to electron-deficient radicals [19]. Therefore, we speculate that the deletion of the C-terminal amino acid may be the reason the antioxidant activity of the modified peptide is not sensitive to the electric field intensity. The immunomodulatory activity of the peptide was further assessed in subsequent experiments to assess whether VNAVLH has substantial immunological activity and to assess whether the immunologically active site of the original peptide treated with PEF is also on the C-terminal amino acid.

### 3.2. Immunomodulatory Activity of Peptides in RAW 264.7

By using the MTT test, the cytotoxic effects of peptides on cells were assessed in order to find the non-toxic concentration. In Figure 2A, it was discovered that neither of the two peptides exhibited any cytotoxic effects on RAW 264.7 cells at concentrations between 31.25 μg/mL and 1.00 mg/mL compared to the control group. Therefore, the concentration of 1.00 mg/mL was used to investigate the cellular immune activity of peptides.

The primary and essential phase in the immunological response is macrophage phagocytosis [32]. Increased phagocytosis is one of the most noticeable outcomes of macrophage activation [33]. Therefore, the phagocytic capacity of RAW 264.7 cells was calculated as a measure of the immune function activation using a neutral red uptake test. As shown in Figure 2B, the peptide VNAVL significantly (*p* < 0.05) improved the phagocytose ability of macrophages, but it was significantly (*p* < 0.05) less effective than VNAVLH. Moreover, the phagocytic ability of VNAVLH was significantly increased from 30.14 ± 1.35% to 45.14 ± 3.23% with the enhancement of field strength and reached the highest value at 40 kV/cm (*p* < 0.05), while VNAVL did not change significantly before and after PEF treatment (*p* > 0.05). After the removal of the terminal histidine, the effect of the original peptide on the phagocytic ability of macrophages was significantly reduced, and the effect of PEF on the original peptide was not reflected on the modified peptide. The results illuminated that C-terminal histidine may be in a critical position for the impact of PEF treatment on the immunological activity of the original peptide and may play a substantial role in the phagocytosis of macrophages during the immune response.

Nitric oxide (NO), which is present in many tissues and organs of living things, plays a crucial role in physiological pathological processes [34]. In the current research, the stimulatory impacts of peptides on NO generation were depicted in Figure 2C. Both VNAVLH and VNAVL significantly (*p* < 0.05) increased NO generation in RAW 264.7 cells relative to the control group, though less than LPS. However, the NO production of macrophages under the action of VNAVL was significantly lower than that of VNAVLH (*p* < 0.05). It is worth noting that NO production of RAW 264.7 cells in PEF-treated VNAVLH significantly increased with field strength and reached the highest value at 40 kV/cm (21.49 ± 0.29 μM), while there was no significant difference between the untreated and PEF-treated VNAVL (*p* > 0.05). This trend was similar to the results of the effect of the peptides on phagocytosis capacity of RAW 264.7 cells. It indicated that PEF can maximize the immune activity of VNAVLH at 40 kV/cm, while VNAVL has no immune activity. Therefore, the electric field intensity of 40 kV/cm was chosen as the PEF treatment condition in subsequent experiments to explore the effect of PEF treatment on the release of cytokines from macrophages.

Macrophages are essential for the function of immune system because they may destroy infections both directly through phagocytosis and indirectly through the release of cytotoxic chemicals such as NO or cytokines such as TNF-α and IL-6 [35]. The effects of PEF-treated and untreated peptides on levels of TNF-α and IL-6 in RAW 264.7 cells were analyzed and are presented in Figure 2D,E. The cytokine production of RAW 264.7 cells in VNAVLH was significantly higher than in the control. Contrary to the results depicted in Figure 2B,C, the production of cytokines by RAW 264.7 cells incubated in the presence of VNAVL was not significantly different from the control. It indicated that VNAVLH can exhibit prominent immune activity by stimulating macrophages to release cytokines. Although VNAVL was able to stimulate the phagocytosis of macrophages, it did not further release cytokines. TNF-α and IL-6 concentrations following treatment with PEF-treated VNAVLH were significantly (*p* < 0.05) enhanced by 1538 pg/mL and 230 pg/mL, compared to the untreated peptides, but those after treatment with PEF-treated VNAVL were not significantly changed (*p* > 0.05). It demonstrated that PEF can further enhance the effect of VNAVLH on the release of cytokines from macrophages but cannot activate the ability of macrophages to release cytokines under the action of VNAVL.

According to the results of immunological activity, the original peptide could stimulate RAW 264.7 cells to produce and secrete TNF-, IL-6, and NO, and the activity of the original peptide could be enhanced after PEF treatment. However, the modified peptide was not immunologically active, and PEF treatment could not trigger its immunological activity. It indicated that the C-terminal histidine of the original peptide might be the key amino acid that leads to its immunological activity and the action of PEF treatment. In subsequent tests, we looked at the secondary structural alterations of the original peptide and the C-terminal-modified peptide in order to shine a light on the underlying mechanism through which PEF treatment results in changes in peptide activity.

### 3.3. Effects of PEF Treatment on Secondary Structure of Peptides

The primary use of circular dichroism (CD) spectroscopy is to identify the secondary structure of peptides [36]. Figure 3 depicts the impact of PEF treatment on the secondary structure of peptides. The positive peaks at 195 nm and 225 nm were seen in the CD spectra of the original peptide as well as the C-terminally changed peptide, as shown in Figure 3. After PEF treatment, the peak intensities of both peptides were stronger at 195 nm, while the change in VNAVL is not prominent. The secondary structure contents of the peptides are depicted in Table 1. The secondary structure of VNAVLH consists of an α-helix, β-sheet, and random coil, whereas VNAVL has no α-helix in comparison, which may be due to the sequence of the original peptide shortened after removing the C-terminal amino acid in the original peptide. Compared with the untreated peptides, the β-sheet content of VNAVLH and VNAVL were decreased by 7.4 ± 0.05% and 2.2 ± 0.05%, respectively, while the random coil content increased correspondingly (*p* < 0.05). In addition, the α-helix content of VNAVLH also showed a slight increase (*p* < 0.05) after PEF treatment. It indicated that the removal of terminal amino acids can impair the ability of the PEF treatment to convert the original peptide from a β-sheet to a random coil.

Variations in the secondary structure of the protein during the simulations are shown in Figure 4. The aim of DSSP analysis is to measure the number of amino acids involved in the different secondary structures of the peptides as a function of time [37]. The secondary structure of VNAVLH is mainly composed of coils, bends, and turns, while that of VNAVL is mainly composed of coils and turns. As shown in Figure 4A, the secondary structure of the peptide molecule fluctuated throughout the simulation, and significant changes in the DSSP graphic plots were observed in Figure 4B. At 250 ns, a significant fluctuation was visible at 40 kV/cm. It showed that the stability of VNAVLH declined after being treated with PEF, which might be because PEF altered the secondary structures of peptides, which altered the orientation of the partial residues [38]. However, no significant fluctuation was observed in the whole simulation of VNAVL (Figure 4C), which remained unchanged after electric field treatment (Figure 4D). The number of amino acids involved in various peptide structures and any changes in those numbers following PEF treatment was revealed by DSSP analysis of the peptide structures (Table 2). It indicated the turn structure of VNAVLH consists of 0.100 amino acids, which decreased by 0.052 amino acids following PEF treatment. Correspondingly, the number of amino acids involved in bend and coil structures rose by 0.008 and 0.044, respectively. Meanwhile, the secondary structure of VNAVL only involved the addition of 0.006 residues to the coil structure, which was consistent with the secondary structure analysis of CD.

According to CD spectra and DSSP analysis, the PEF treatment can change the secondary structure of the peptides to a random coil, revealing the active site and fully enabling biological activities. However, the effect of this change on VNAVL is far less than that of VNAVLH, which was consistent with the findings of the activity tests. Therefore, we speculate that PEF treatment exposes the key active site located on the C-terminal histidine by altering the secondary structure of the peptide.

## 4. Conclusions

The aim of this study was to compare the effect of PEF treatment on the structure–activity relationship of the original peptide (VNAVLH) and C-terminal-modified peptide (VNAVL) and identify the key active residue for PEF action. Both the original and modified peptides had antioxidant activity, and the modified peptide’s antioxidant activity was much lower than that of the original peptide. However, the modified peptide was not immunologically active, and PEF treatment could not trigger its immunological activity. In addition, the antioxidant and immunomodulatory activities of the original peptide were significantly more sensitive to treatment with 40 kV/cm than other field intensities, which was not the case for the C-terminally modified peptide. CD spectra and DSSP analysis revealed that the PEF treatment could transform the β structure to a random coil in the peptides’ molecule, leading to revealing the active site and fully playing biological activities. Moreover, the conversion ability of the C-terminal-modified peptide is far less than that of the original peptide. Therefore, C-terminal histidine of the original peptide might be the key amino acid that leads to its biological activity, and PEF treatment exposes the key active site located on the C-terminal histidine by altering the secondary structure of the peptide. This study will contribute to the application of PEF technology in the development of more effective novel antioxidant and immunomodulatory peptides.

## Figures and Tables

**Figure 1 foods-11-02649-f001:**
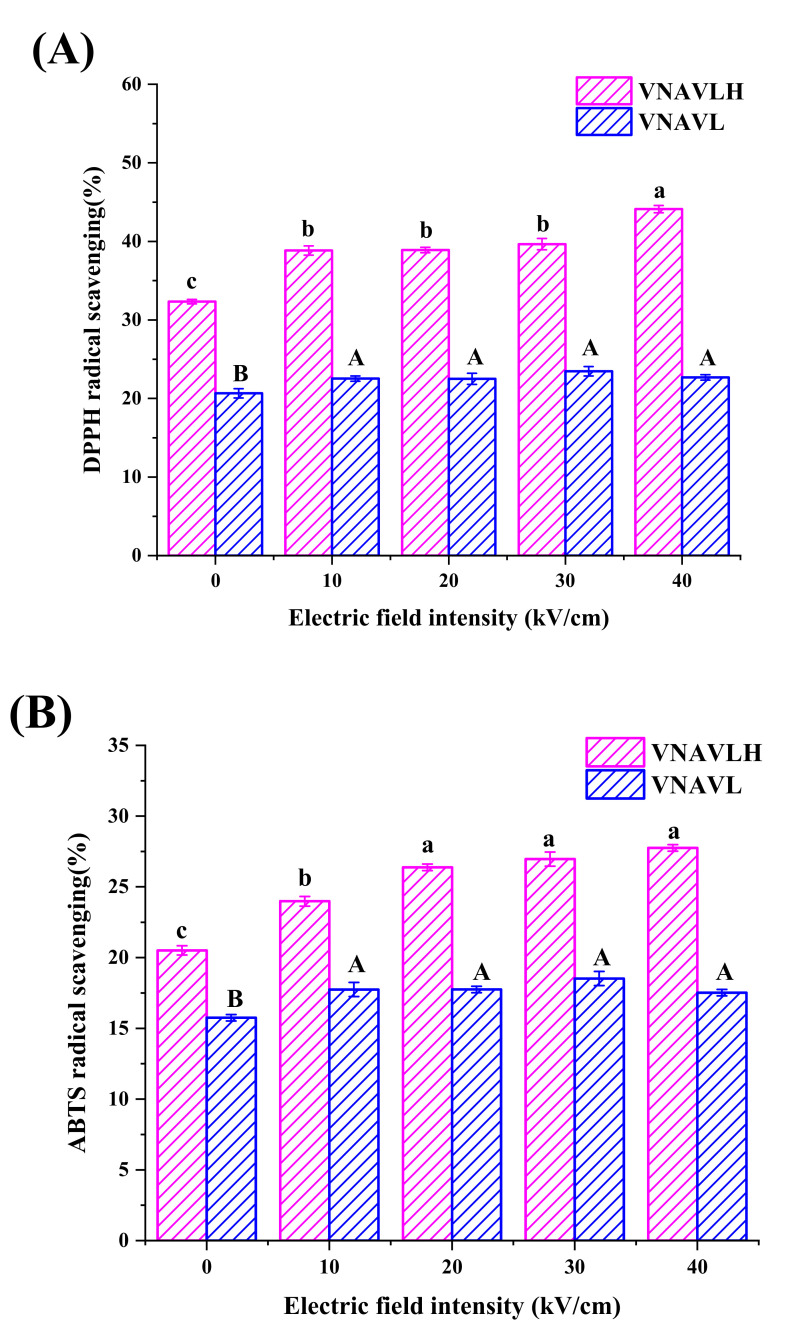
Effects of PEF treatment on (**A**) DPPH and (**B**) ABTS radical scavenging of peptides at different electric field intensities of 10, 20, 30, and 40 kV/cm, respectively. All data were expressed as mean ± standard deviation (*n* = 6). The different lowercase letters represent significant differences between different electric field intensities of VNAVLH (*p* < 0.05). The different capital letters represent significant differences between different electric field intensities of VNAVL (*p* < 0.05).

**Figure 2 foods-11-02649-f002:**
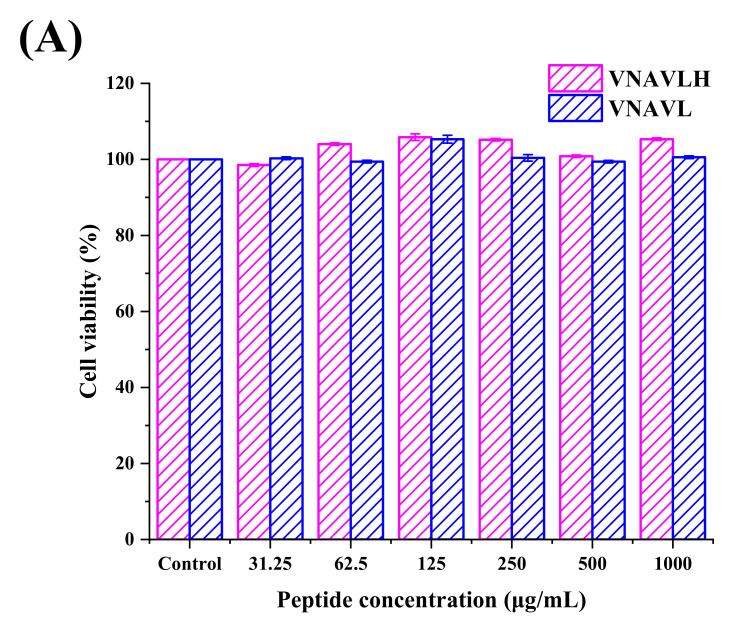
Effects of PEF treatment on (**A**) cell viability; (**B**) phagocytosis; (**C**) production of NO; (**D**) TNF-α; and (**E**) IL-6 at the electric field intensities of 10, 20, 30, and 40 kV/cm, respectively. All data were expressed as mean ± standard deviation (*n* = 6). The different lowercase letters represent significant differences between different electric field intensities of VNAVLH (*p* < 0.05). The different capital letters represent significant differences between different electric field intensities of VNAVL (*p* < 0.05). The asterisk indicates significant differences between different peptides (*p* < 0.05).

**Figure 3 foods-11-02649-f003:**
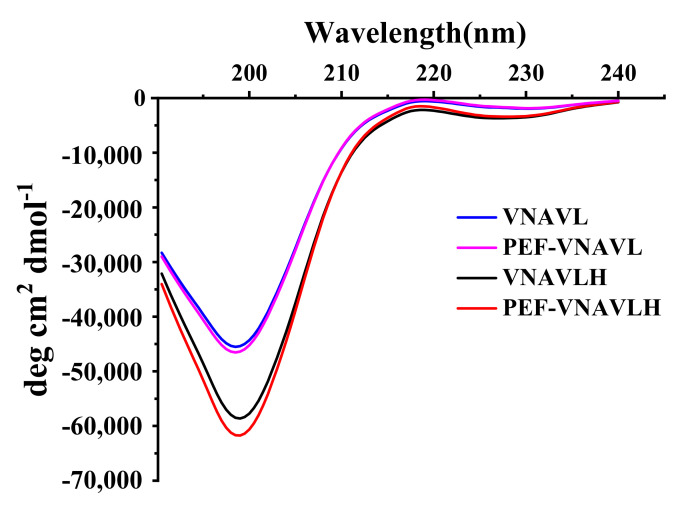
CD spectra of peptides before and after PEF treatment.

**Figure 4 foods-11-02649-f004:**
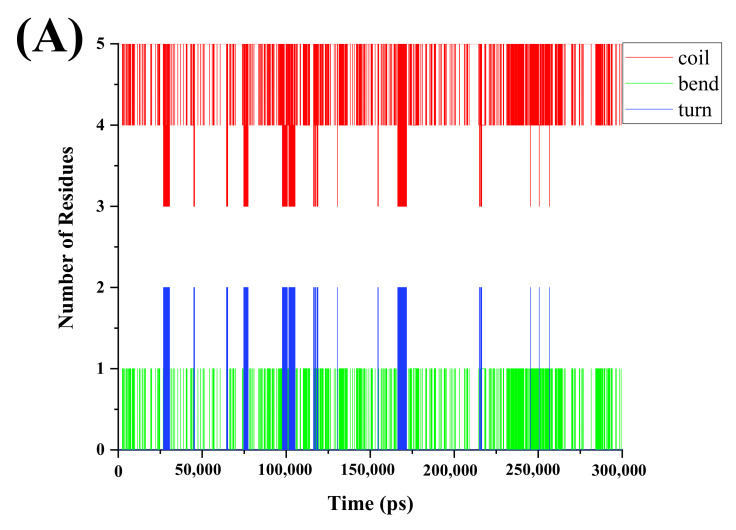
The DSSP graphic plots of (**A**) untreated VNAVLH, (**B**) PEF-treated VNAVLH, (**C**) untreated VNAVL, and (**D**) PEF-treated VNAVL, respectively.

**Table 1 foods-11-02649-t001:** The content changes of secondary structure of peptides before and after PEF treatment.

Secondary Structure	VNAVLH (%)	PEF-VNAVLH (%)	VNAVL (%)	PEF-VNAVL (%)
α-Helix	2.7 ± 0.1	4.7 ± 0.1	-	-
β-sheet	19.9 ± 0.15	12.5 ± 0.2	35.1 ± 0.25	32.9 ± 0.2
Random coil	77.4 ± 0.25	82.8 ± 0.3	64.9 ± 0.25	67.1 ± 0.2

All data were expressed as mean ± standard deviation (*n* = 3).

**Table 2 foods-11-02649-t002:** The percentages of average secondary structure for peptides during the 300 nsec MD simulation.

Secondary Structure	VNAVLH	PEF- VNAVLH	Difference Amount	VNAVL	PEF- VNAVL	Difference Amount
Coil	4.703	4.747	0.044↑	3.984	3.990	0.006↑
Bend	0.196	0.204	0.008↑	-	-	-
Turn	0.100	0.048	0.052↓	0.015	0.009	0.006↓

The values of DSSP are given in numbers. DSSP (database of secondary structure assignments for all proteins). The up/down arrows indicate the rise/fall of the secondary structure of the peptides after PEF treatment.

## Data Availability

The data showed in this study are contained within the article.

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
