# Peer review of "C-Terminal Modification on the Immunomodulatory Activity, Antioxidant Activity, and Structure–Activity Relationship of Pulsed Electric Field (PEF)-Treated Pine Nut Peptide"

_foods, 2022, doi:10.3390/foods11172649_

Round 1

Reviewer 1 Report

The manuscript of Zhang et al. describes the effect of the treatment with pulsed electric fields (PEF) on two peptides (VNAVLH and VNAVL) deriving from pine nut. The immunomodulation capacity on specific phagocytic cells of one peptide  (VNAVLH) is clear and dependent on the treatment intensity, the other peptide seems not to be capable of eliciting significant effect.

The authors try to discuss these results, suggesting possible structural modifications induced by PEF treatment.

In my opinion, this manuscript is not suitable for Foods, but rather for a more pharma-oriented manuscript. It is not possible to understand where these peptides where originated from (digestion?), and the reference provided by the authors N. 12)  does refer to another peptide characterized in the same way (RGAVLH).

There are several typos and inaccuracies. For instance:

-          The Maillard reaction does not bring to glycosylation, but to glycation (line 39)

-   No production increased by 6.26 uM (line 53): here I would mention a %, not a concentration)

-          - “results illuminated that…” (line 178): not an English form;

-          - 100 mL of DPPH (line95): I guess 100 uL

-          - celllysate (1 M anhydrous ethanol: 1:1, v/v) (line 119): I don't understand what's in brackets.

The phagocytosis assay is not clearly described, while on the contrary, its rationale should be clearly explained as it is a central experiment.

Looking at the CD spectra (fig. 3A) it is clear that these peptide do not possess a structure. The minima at 200 nm are typical of random coil structures. So I wonder what the following discussion about the different secondary structures is based on since there are no structures… It is difficult, if not impossible that such small peptides of few amino acids possess a structure.

Reviewer 2 Report

Manuscript Title:
The effect of C-terminal modification on the immunomodulatory activity, antioxidant activity and structure-activity relationship of pine nut peptide treated by pulsed electric field (PEF) treatment
Comments:
1. Please consider rewriting the abstract for clarification.
2. Electric field intensities were set from 10 kV/cm to 40 kV/cm; why this range was chosen?
3. Experimental design: how many independent replications were carried out in the study?
4. Statistical analysis: the results were all presented as means with standard deviations (SD) but n=??
5. The link between increased field intensities and biological activities in the original as well as in the modified peptide needs further discussion.
6. Line 185: the previous study needs to be cited if published.
7. Figure 2 (A): was any statistical analysis conducted?
8. Please explain (as figure notes) what different letters (upper/lower case) depict in figure 2.
9. The manuscript has writing errors. Therefore, the whole manuscript must be re-revised for clarity, and grammatical and writing errors.

Reviewer 3 Report

This manuscript is on the analysis of the potential influences of PEF treatments on the structure-function characteristics of a certain pine nut peptide and its modified histidine bearing counterpart.

The findings are of relevance to the Journal`s readers. The technical merit of the manuscript is good. The interpretation of data is generally good. The Introduction section may be improved. The use of English language is mostly good.

 Consequently, I would like to recommend minor revisions.

I think that the Title is understandable but may be improved. Please consider this point.

Introduction. What is the operational basis behind PEEF and what is the mechanism by which you would anticipate a change in your samples? Please clarify.

L85. …the peptide Peptides? I do not think that this expression is practical.

Throughout the text, please reduce the usage of `s or s`. I tend to think that most authors minimally use these in scientific papers.

L226: The production cytokines?

L217: Unclear. Please paraphrase.

What are the potential uses/implications for your findings? Please comment.

Personal opininon (not critical). I think that Figure 3B data would be more useful as a table.

L322:… acid

Round 2

Reviewer 1 Report

The authors improved the quality of the work by better describing the methods and providing a more in-depth discussion of the results. Unfortunately, the results of the study of the secondary structure still do not convince me. The structure of the peptides is markedly random coil. I would not rely on the results of a predicting software based on evidently random coil structures. Five or 6 amino acids are insufficient to define a beta-sheet. For this reason, I would not speak of beta sheets but rather of beta strands. The quality of the English language and style must be improved. In particular, the clarity and the form of the abstract are still inadequate.